# Thermal Conductance of Graphene-Titanium Interface: A Molecular Simulation

**DOI:** 10.3390/molecules27030905

**Published:** 2022-01-28

**Authors:** Bingxian Ou, Junxia Yan, Qinsheng Wang, Lixin Lu

**Affiliations:** 1School of Mechanical Engineering, Jiangnan University, Wuxi 214122, China; obxobx@163.com (B.O.); yjx@jiangnan.edu.cn (J.Y.); 2Special Equipment Safety Supervision Inspection Institute of Jiangsu Province, National Graphene Products Quality Inspection and Testing Center (Jiangsu), Wuxi 214174, China; wqs@wxtjy.com

**Keywords:** thermal boundary conductance, titanium-graphene, molecular dynamics

## Abstract

Titanium is a commonly used material in aviation, aerospace, and military applications, due to the outstanding mechanical properties of titanium and its alloys. However, its relatively low thermal conductivity restricts its extended usage. The use of graphene as a filler shows great potential for the enhancement of thermal conductivity in titanium-based metal-matrix composites (MMCs). We used classical molecular dynamics (MD) simulation methods to explore the thermal conductance at the titanium–graphene (Ti/Gr) interface for its thermal boundary conductance, which plays an important role in the thermal properties of Ti-based MMCs. The effects of system size, layer number, temperature, and strain were considered. The results show that the thermal boundary conductance (TBC) decreases with an increasing layer number and reaches a plateau at n = 5. TBC falls under tensile strain and, in turn, it grows with compressive strain. The variation of TBC is explained qualitatively by the interfacial atomic vibration coupling factor. Our findings also provide insights into ways to optimize future thermal management based on Ti-based MMCs materials.

## 1. Introduction

Compared with other metals, titanium and its alloy derivatives are known for their excellent physical properties, stable chemical properties, and great biocompatibilities and are widely applied in aerospace [1], the military [2], automobile manufacturing [3], bone substitute manufacturing [4,5], and other fields. Titanium has a high strength-to-mass ratio. For example, titanium is 60% denser than aluminum but more than twice as strong as the most commonly used aluminum alloy [6]. The tensile strength of some titanium alloys can even reach 1400 MPa [1,7]. The relatively high melting point (more than 1650 °C) makes titanium useful as a refractory metal. In addition, titanium and its alloys have good corrosion resistance [8,9]. However, the vital drawbacks of titanium and its alloys cannot be ignored. It loses strength when heated above 430 °C and is a poor conductor of heat and electricity. How to improve the thermal conductivity of titanium under the condition of ensuring the mechanical properties of titanium has become a key issue in engineering application research.

Due to outstanding properties, such as high thermal conductivity (~3000 Wm^−1^ K^−1^) [10,11], mechanical strength (~125 GPa) [12,13], and tensile modulus (~1.1 TPa) [14], graphene–metal materials exhibit high performance in thermal and mechanical properties and enhance the chemical stability of the metal [15,16,17,18,19,20]. For example, few-layer graphene (FLG)-reinforced copper composites were fabricated by spark plasma sintering (SPS) with an FLG volume fraction of 2.4 vol% [13]. The conductance of composites reached 70.4% of international annealed copper standard (IACS) [21,22]. Graphene-Cu nanocomposites foils were synthesized by an electrochemical method showed a high level of hardness, up to 2.2–2.5 GPa, and an elastic modulus of 137 GPa, which were increased by 96% and 30% from those of pure Cu, respectively [12,23]. Graphene-Al composites were obtained by powder a metallurgy technique and exhibited greatly improved mechanical properties after adding graphene [7,24,25]. Therefore, the thermoelectric properties and mechanical stability of pure Ti and its alloys may be effectively enhanced by adding a certain amount of graphene. 

The thermal conductivity of Ti-based metal-matrix composites increases with amount of graphene, as confirmed through the experiments in the previous studies [19,26,27,28]. Gr/Ti composites outperform most carbon nanotube/Ti in thermal conductivity. Furthermore, the thermal conductivity and specific heat capacity of the composite material increase drastically with the increase of the graphene content [29,30]. Zheng et al. [31] reported that when 8 atomic layers of graphene were affiliated to the Gr/Ti composites, the interface thermal conductivity was as high as 440 MW/m^2^ K. However, there is still a vacuum in the research on the physical mechanism behind this phenomenon.

In this work, we study the thermal boundary conductance at the Ti/Gr interface using both non-equilibrium molecular dynamics (NEMD) and thermal relaxation (TR) methods. The NEMD simulation is helpful in understanding the thermal conductivity of the composite and the TR method mimics the experimental laser-based pump-probe process. The relationship between the thermal boundary conductance and the phonon spectra at the interface is discussed in great detail. Our work suggests that the thermal properties of titanium can be greatly improved by affiliating an appropriate amount of graphene. In addition, compressive strain is an effective method to enhance the TBC of the Ti/Gr interface. 

## 2. Materials and Methods

Ti is a kind of metal with hexagonal closest packed (HCP) structure. We chose three of the most common crystal faces, Ti(10-10), Ti(11-20), and Ti(0001), as shown in Figure 1a, to build the model of Ti/Gr interfaces. In order to establish the Ti/Gr interface structure with minimal lattice mismatch, we selected different lengths for the titanium and graphene models for the corresponding crystal faces. Table 1 lists the dimensions of the models in the x-direction and y-direction, as well as the strain rate of graphene in each direction, where graphene sheets were subjected to a tensile strain at both the x- and y-directions with Ti blocks keeping strain-free. The thickness of the titanium in the z-direction was about 2 nm.

All molecular dynamics (MD) simulations were performed using the large-scale atomic/molecular massively parallel simulator (LAMMPS) package [32]. The modified embedded-atom method (MEAM) potential [33] was employed to describe the atomic interactions in titanium, while the interactions between carbon atoms were described by the optimized Tersoff potential [34]. The van der Waals interactions of interlayers were modeled with Lennard–Jones (LJ) function $V\left(r_{ij}\;\right)=4\varepsilon \left[{\left(\sigma /r_{ij}\right)}^{12}-{\left(\sigma /r_{ij}\right)}^{6}\;\right]$ and the corresponding parameters are shown as σ_Ti−Gr_ = 0.36 nm and ε_Ti−Gr_ = 0.006535 eV [35]. The time step was set as 0.5 fs. The cutoff distance of the LJ potential was set as 1.2 nm. The non-periodic boundary conditions (Figure 1b,c) were employed to explore the TBC at the Ti/Gr interface for both the NEMD and thermal relaxation methods.

### 2.1. NEMD Method

To calculate the TBC based on the NEMD, simulations were conducted using the nonperiodic boundary condition model (Figure 1b), in which the periodic boundary conditions were applied along the in-plane (x and y) directions, with the free-boundary condition along the out-of-plane (z) direction. We set the heat zone and the cold zone at the temperatures of 340 K and 260 K by using the Langevin thermostat [36]. The system occurred a directional thermal transfer process because of the temperature difference. After 0.5 ns, a temperature gradient was generated and each part was time-independent. Any block of the Ti was divided into 25 equal slabs along the heat flux direction (z) to calculate the temperature distribution, and the last 3 ns of data was collected. Heat flux is the change in energy over time: J = dΔE(t)/dt, where ΔE represents the energy change. TBC can be calculated as follows: G = J/AΔT, where A is the cross-section of heat transfer. Herein, we take the value of ΔT by averaging both of the two-temperature differences that occurred close to the heat source and the heat sink of the Ti/Gr interfaces. 

### 2.2. Thermal Relaxation Method

In the TR studies, the non-periodic boundary condition model in the surface-normal direction, as shown in Figure 1c, was employed. A time step of 0.5 fs was used in the MD simulations. First, the initial configuration was equilibrated at 300 K by performing a constant volume and a constant temperature (NVT) for 1.0 ns (2 × 10^6^ steps) to relax the structure of the system and to optimize the interlayer distance. Afterwards, the system was then switched to the microcanonical ensemble (NVE). The initial temperature gap could be generated by increasing the temperature of the graphene layers instantaneously to a specified value by rescaling the velocities of the carbon atoms [37]. After the heat source was removed, the temperature of the graphene layer decayed exponentially within a few hundred picoseconds. The temperature profile of the graphene, as shown in Figure 2, was fitted by an exponential function and the relaxation time τ was obtained using $\Delta T=\Delta T\left(t_{0}\right)\exp\left(\left(t_{0}-t\right)/\tau \right)$. Then, the value of the TBC was calculated via G=C/(A×τ), where C is the heat capacity of graphene. The heat capacity, C, of graphene was computed, using the molecular dynamics approach, as a function of temperature [38,39]. The calculation was performed by varying the temperature in 5 K increments over a ±10 K range around the temperature of interest. For instance, the total energy, E, was computed at T = 290, 295, 300, 305, and 310 K. The results were averaged over 1.5 × 10^5^ steps under five independent simulations. The slope of a linear fitting is the heat capacity at the interest temperature. 

### 2.3. Phonon Spectra 

To elucidate the heat transfer mechanism across the interface, we investigated the vibrational density of states (VDOS) at the interfaces obtained from the Fourier transform of the interfacial atoms’ velocity autocorrelation [40].
(1)$$P\left(\omega \right)=\frac{1}{\sqrt{2\pi }}\int_{0}^{\infty }e^{i\omega t}<\overset{N}{\underset{j=1}{\sum }}v_{j}\left(t\right)v_{j}\left(0\right)>dt$$

The thermal transport between graphene and Ti is mainly dominated by the phonon in the out-of-plane direction, and most of the phonon coupling is contributed by the low-frequency zone [41]. The overlap of the phonon spectrum parameter, S, is defined as: (2)$$\begin{array}{c} S=\frac{\int_{0}^{\infty }P_{Gr}\left(\omega \right)P_{Ti}\left(\omega \right)d\omega }{\int_{0}^{\infty }P_{Gr}\left(\omega \right)d\omega \int_{0}^{\infty }P_{Ti}\left(\omega \right)d\omega }\; \end{array}$$
where $P_{Gr}\left(\omega \right)$ and $P_{Ti}\left(\omega \right)$ denote the phonon spectra at frequency ω of carbon and Ti atoms at the interface, respectively.

## 3. Results and Discussion

### 3.1. Thermal Boundary Conductance at the Ti/Gr Interface

We first compared the TBC results obtained from both two methods, as shown in Figure 3. The TBC of Ti(10-10)/Gr exhibited the worst performance and Ti(0001)/Gr showed the best, as illustrated in Figure 3a. Here we normalized the TBC value of Ti(0001)-graphene interface as 1.0 to obtain the tendency of the TBC. The results show that the TBC obtained by the two methods was almost completely consistent with the variety of the crystal faces. The (0001) surface had the highest TBC, followed by the (11-20) surface. The (10-10) surface had the lowest TBC, indicating the robustness of the simulations. Although, the TBC of NEMD about the Ti(0001)-graphene interface was 425 MW/m^2^ K, while the TBC of the thermal relaxation methods was 161 MW/m^2^ K. To elucidate the inner mechanism of the TBC diversity obtained from both the NEMD and TR methods, we calculated the phonon spectrum and the phonon coupling parameter, as shown in Figure 3b,c. The phonons in Figure 3b mainly distributed in the range of 1–30 THz, while Figure 3c shows that phonons only existed in the interval of 1–40 THz. The NEMD method can form a stable heat source and temperature gradient, including a higher frequency of phonons participating in interface heat conduction. For the thermal relaxation method, the initial high temperature brings about mostly high-frequency phonons, while the low-frequency phonons play a major role in the TBC, which results in the thermal relaxation method providing lower measurements than the NEMD method. This is also consistent with the previously reported results [42,43].

In addition, we analyzed the relationship between the phonon coupling strength and the TBC of different crystal faces. Since the results of NEMD were consistent with the thermal relaxation method, here we computed the TBC with different crystal faces using the TR method in Figure 4. The phonon spectra of the different crystal faces are illustrated in Figure 4a–c. The distribution range of phonons is almost the same, with only a slight difference in intensity in the low-frequency coupling range, which is the main reason for the diversity in the TBC (Figure 4d). Therefore, the choice of different crystal faces has a prominent influence on the heat transfer effect.

### 3.2. Temperature Effects

We further explored the dependence of G on the environmental temperature, T, using the NEMD method for precise temperature maintenance and easy control. As shown in Figure 5, the TBC gradually increases with temperature, from 409 MW/m^2^ K at 250 K to 502 MW/m^2^ K at 450 K. It features an almost linear dependence relationship (the fitted linear equation is: G = aT + b, here a = 0.48 and b = 287.36) for the Umklapp processes, which play the dominant role in heat conduction [44]. Correspondingly, the overlapping factor, S, also increases linearly with temperature. Furthermore, the representative phonon spectra are illustrated in Figure 6 to obtain insight into the heat transfer mechanism of the TBC of Ti/Gr interface.

Comparing the in-plane and out-of-plane phonon spectra with the different temperatures in Figure 6, there is an interesting phenomenon that both the in-plane and out-of-plane phonon spectra of Ti had almost the same frequency and peak distributions, while the graphene had an enormous difference. This is attributed to the fact that graphene is a two-dimensional material, exhibiting enormous anisotropic properties between the in-plane and out-of-plane directions. Since the phonon spectra of Ti were almost all distributed in the low frequency region (1–10 THz), out-of-plane phonons dominated the TBC, and its overlapped parameter S was about one order higher than that of the in-plane phonon overlap S value. The phonon coupling strength was enhanced at the out-of-plane components, and the corresponding S values increased from 0.0318 to 0.0376 with the temperature. 

### 3.3. Size Effect of Graphene

We studied the TBC variation with graphene layers, N. The layer number of a graphene sheet can be controlled by CVD growth or physical translation. Here, the interfacial model of Ti(0001)/Gr model was employed and n represents the number of graphene layers, with the consideration of a few layers of graphene stacking in the experiment. The simulations were carried out by the NEMD method at 300 K, and the corresponding schematic model is shown in Figure 7. Both ends were fixed with cold (heat sink) and hot (heat source) regions neighboring fixed regions. As shown in Figure 7b, the TBC decreased with an increasing n until it converged at n = 5. There was a reduction of 85% from 425 ± 11 MW/m^2^ K (n = 1) to 63 ± 1 MW/m^2^ K (n = 5). This was due to the increasing thickness of graphene block, which enhanced the interfacial phonon scattering at the interface. Similar phenomena have been reported in heat transport across the graphene/Cu and graphene/Ni interfaces [45,46]. To explain this phenomenon, the corresponding in-plane phonon and out-of-plane coupling strength are illustrated in Figure 7c,d, respectively. The in-plane phonon coupling strength were almost independent of n. It was, apparently, not the reason for the reduction of the TBC. The out-of-plane phonon coupling strength monotonically decreased with the number of layers, and the corresponding overlapping factor S dropped from 0.034 (n = 1) to 0.020 (n = 9), which is consistent with the trend of the TBC. Figure 7 shows that the magnitude of overlap factor S in the out-of-plane direction was about one order higher than that of the in-plane direction. From the analysis of the previous results, the out-of-plane phonons of graphene play a dominant role on the TBC.

### 3.4. Effect of Strain Engineering

The thermal conductivity and the TBC are sensitive to the conducted strain. Applying a cross-face strain is an effective method to tune the heat transport across the interface, as confirmed in the previous studies of graphene [47], MoS_2_ [48], and black phosphorus [49]. Here, we performed a surface-normal strain along the direction of the TBC using the NEMD model. The uniaxial strain was applied by adjusting the d (distance) between two bulks of Ti. As illustrated in Figure 8a, the strain changes from tension to compression (3–−3%). The graphene sheet sandwiched between two Ti bulks transformed from a state full of ripples to a flat state. The stress-strain relationship and the distance between the two bulks of Ti, d, with conducted strain are depicted in Figure 8b. When the strain was about −6%, the stress was approximately 120 GPa and the graphene structure broke at such a high pressure. Thus, we only studied the strain range from −3% to 3%. The block interlayer distance, d, increased monotonously with increasing strain in Figure 8b. 

The compressive strain was found to enhance the thermal conductance, G, and followed an exponential dependence relationship, which is consistent with previous studies [42,48]. In Figure 8c, when the strain rate was –3%, the TBC was about 700 MW/m^2^ K, which is almost twice as high as the TBC in the strain-free state. However, the TBC was only about 48 MW/m^2^ K with the strain at 3%. The enormous TBC difference indicates that applying compressive strain to Ti and graphene is an efficient way to improve the TBC. In order to elucidate the strain effect on the TBC, we investigated the variation of phonon coupling strength under different strains. As shown in Figure 8c, the S was about 0.0299 with the compressive strain rate at –3%, while it was about 0.0173 with the tensile strain rate at 3%. There was a satisfying agreement between the overlapping factor, S, and the interface thermal conductivity with the strain rate. These findings are consistent with the above conclusions. Hence, the TBC can be effectively modulated by cross-face strain. The thermal conductivity of Ti/Gr MMCs is computed taking into account TBC effects at the Ti/Gr interfaces, and the thermal conductivity increases with graphene volume content. Our findings also provide insights into ways to optimize future thermal management based on MMC materials.

## 4. Conclusions

In summary, the thermal boundary conductance at Ti/Gr interfaces was investigated systematically via MD simulations. Among the Ti(0001)/Gr, Ti(10-10)/Gr, and Ti(11-20)/Gr models, the Ti(0001)/Gr model exhibited the best performance of TBC at the interface and Ti(10-10)/Gr showed the lowest TBC value. The TBC increased with increasing temperature and its value increased 18.1% when the background temperature raised from 300 K to 450 K. The TBC decreased with increasing interfacial graphene layer numbers. A reduction of 85% was reported for an n-layer (n ≥ 5) graphene block compared with the case of a single-layer graphene interface. Our results exhibit enormous potential of Ti/Gr MMCs in thermal management and cooling applications.

## Figures and Tables

**Figure 1 molecules-27-00905-f001:**
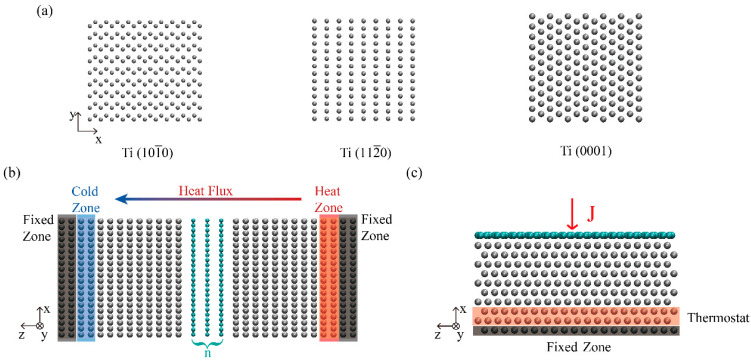
(**a**) Atomic structure of the first layer lattice and structure snapshots of the interfacial Ti layer for the (10-10), (11-20), and (0001) surfaces. (**b**) The schematic model setup for the NEMD method. (**c**) The MD model of the Ti/Gr interface under free boundary condition using the thermal relaxation method.

**Figure 2 molecules-27-00905-f002:**
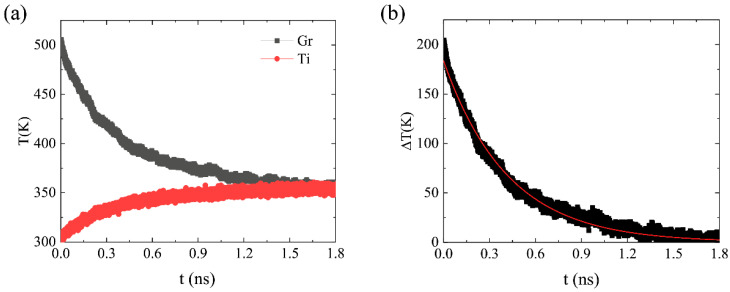
(**a**) Profile of the temperature evolution in the graphene sheets (black line) and the bulk of the Ti (red line) using the thermal relaxation method. (**b**) Profile and exponential fitting curves of ∆T.

**Figure 3 molecules-27-00905-f003:**
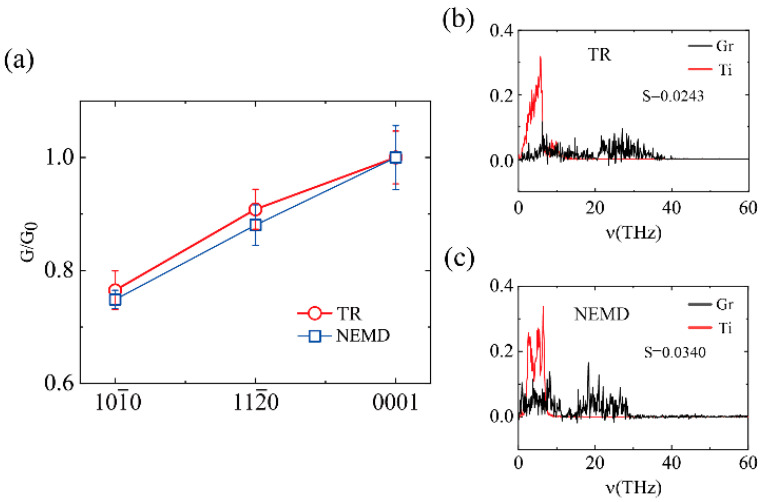
(**a**) The thermal boundary conductance obtained by the two methods, NEMD and TR, varied with the crystal faces of Ti. G_0_ is the TBC value of Ti(0001)/Gr. (**b,c**) The out-of-plane atomic vibration spectra of the Ti(0001)-graphene interface obtained by the NEMD (**b**) and TR (**c**) methods, respectively.

**Figure 4 molecules-27-00905-f004:**
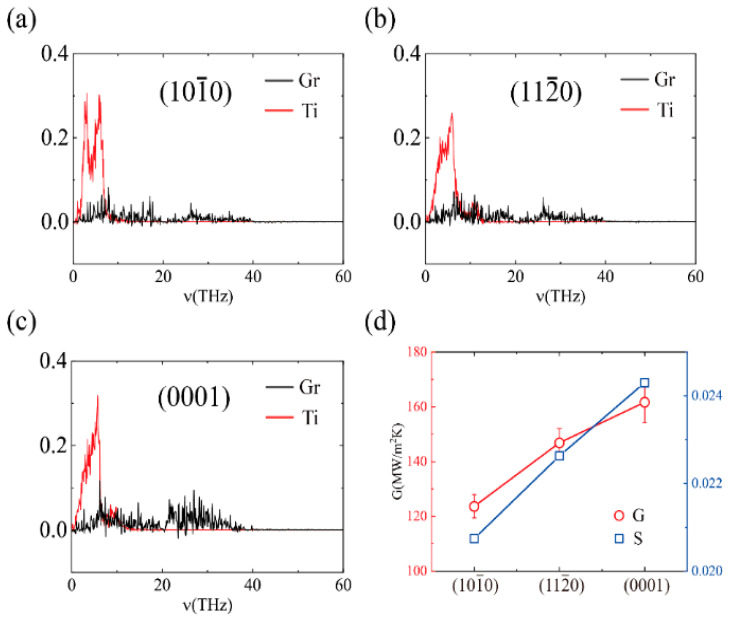
Thermal interface conductance and the out-of-plane phonon spectra of different crystal faces obtained by the heat dissipation method: (**a**) the phonon spectra of the Ti(10-10)-graphene interface; (**b**) the phonon spectra of the Ti(11-20)-graphene interface; (**c**) the phonon spectra of the Ti(0001)-graphene interface; and (**d**) the TBC and phonon coupling strength of the Ti/Gr interface.

**Figure 5 molecules-27-00905-f005:**
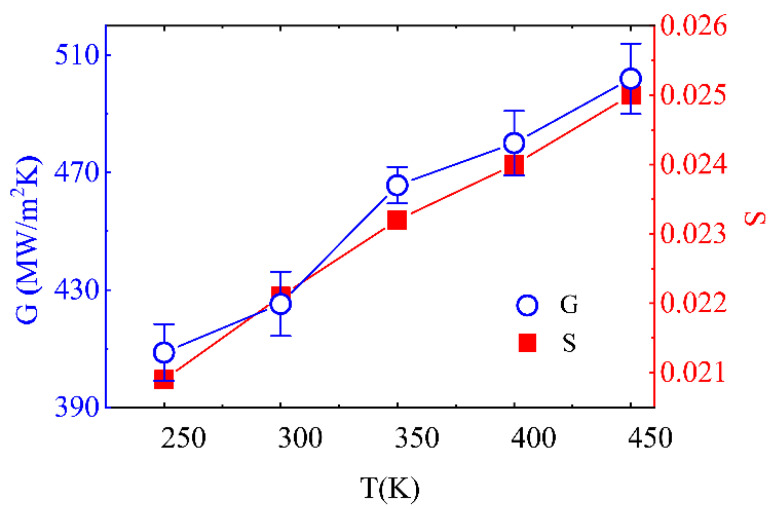
Varying range of the thermal boundary conductance and phonon coupling strength, S, along with temperature.

**Figure 6 molecules-27-00905-f006:**
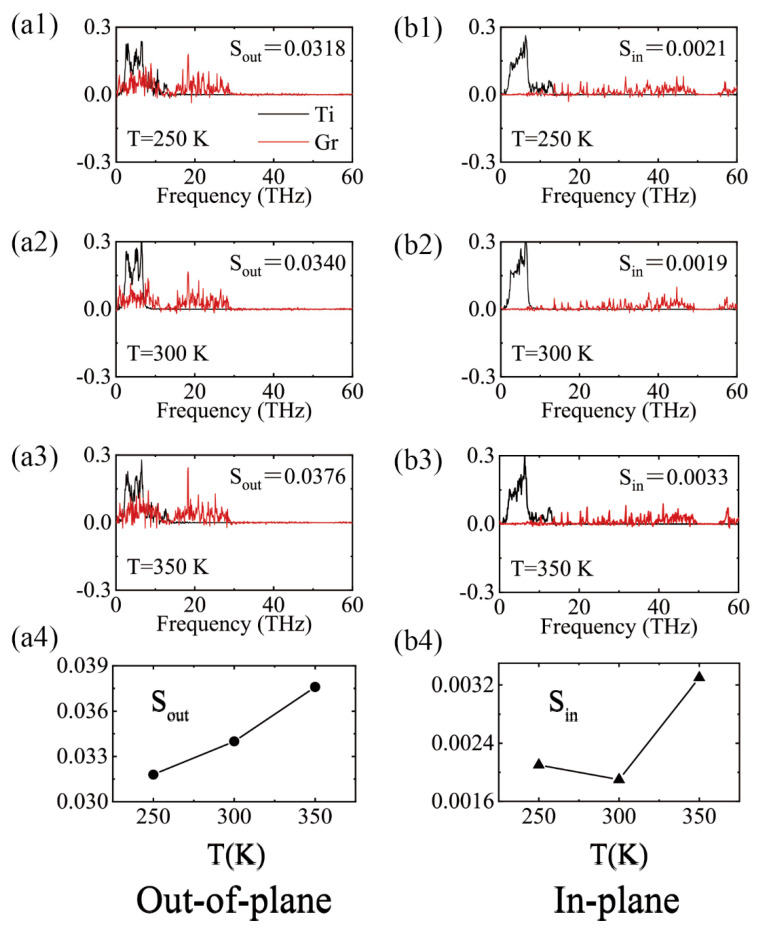
(**a1–a3,b1–b3**) Vibration density of states of atoms in interfaces of Ti and Gr under various temperatures. (**a4**,**b4**) the overlap parameter S of out-of-plane and in-plane dimensions, respectively.

**Figure 7 molecules-27-00905-f007:**
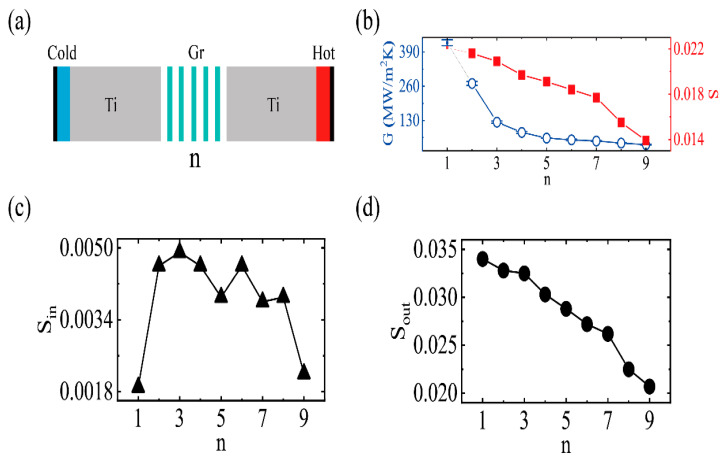
(**a**) The schematic of the thermal boundary conductance with the size effect of graphene. n, represents the number of graphene layers. (**b**) Varying range of the thermal boundary conductance and phonon coupling strength, *S,* along with n. (**c**) Varying range of the in-plane phonon coupling strength, S, along with n. (**d**) Varying range of the out-of-plane phonon coupling strength, S, along with n.

**Figure 8 molecules-27-00905-f008:**
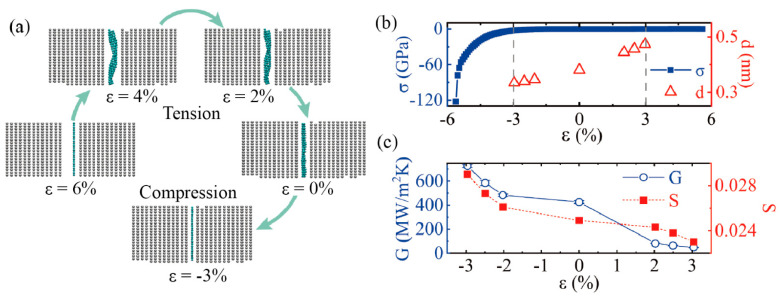
(**a**) The Snapshots of Ti and graphene under different strains. (**b**) Varying range of the stress and distance between two bulks of Ti, along with strain. (**c**) Varying range of the thermal boundary conductance and phonon coupling strength, S, along with strain.

**Table 1 molecules-27-00905-t001:** The simulation parameters of the model with different crystal faces of titanium and the mismatch strain of graphene.

Crystal Face	L_x_ (nm)	ε_Gr-x_	L_y_ (nm)	ε_Gr-y_	L_z_ (nm)
Ti(10-10)	7.2272	0.204%	6.6435	0.0429%	2.0
Ti(11-20)	7.2272	0.204%	5.901	0.0305%	1.98
Ti(0001)	5.1104	0.0310%	5.901	0.0305%	1.959

## Data Availability

The authors can confirm that all relevant data are included in the article.

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
