# Peer review of "Thermal Conductance of Graphene-Titanium Interface: A Molecular Simulation"

_molecules, 2022, doi:10.3390/molecules27030905_

Round 1

Reviewer 1 Report

I found the manuscript very interesting, but I will suggest to move some not crucial (thus basic) equations to the Supplementary - phonon spectra etc.

The same is true for the number of figures - some of them are not important for the whole story of the manuscript and the general reader.

There are wrong references to the figures - few examples below:

*page 3 line 114 - should be Figure 2 ---> is Figure. 4 with dot!

*page 6 line 185 must be Figure 5 not 7

*page 6 line 192 must be Figure 6 not 8

*the is no Figure 10 as wrote on page 8!

*page 2 line 67 after dot with Capital letter

*I am not familiar with the word furthermarle ? Few times in the text

So, I do expect that authors before publications should extensively 

check the language.

Author Response

We greatly appreciate all the reviewers for their professional and constructive comments. We have revised our manuscript carefully based on their comments. In the following, we have addressed the reviewer’s comments point by point. It should be noted that the reviewer’s comments are marked in black italic and the corresponding responses are in blue. All the revisions in the revised manuscript are highlighted in red.

Reviewer #1:

I found the manuscript very interesting, but I will suggest to move some not crucial (thus basic) equations to the Supplementary - phonon spectra etc.

Response: Thanks for your positive comments. The equation of phonon spectra was supplemented in Page 4 line 132.

  1. The same is true for the number of figures - some of them are not important for the whole story of the manuscript and the general reader. There are wrong references to the figures - few examples below:

*page 3 line 114 - should be Figure 2 ---> is Figure. 4 with dot!

*page 6 line 185 must be Figure 5 not 7

*page 6 line 192 must be Figure 6 not 8

*the is no Figure 10 as wrote on page 8!

*page 2 line 67 after dot with Capital letter

Response: Thanks for your comments and pointed out the mistaken labeling. We have modified the miss-labeled Figure Numbers and checked the revised manuscript.

  1. *I am not familiar with the word furthermarle? Few times in the text

Response: Thanks for your comment. It’s a typo and we are sorry for the spelling mistake. The manuscript was checked and typos were modified in the revised manuscript.

In the revised manuscript:

Page 1 line 34: It loses strength when heated above 430°C, furthermore, and is a poor conductor of heat and electricity.

Page 2 line 52: Gr/Ti composites outperform most of carbon nanotube/Ti in thermal conductivity, furthermore, the thermal conductivity and specific heat capacity of the composite material increase drastically with the increase of the graphene content.

Page 6 line 187: Furthermore, the representative phonon spectra are illustrated in Figure 6 to obtain the heat transfer mechanism insight the TBC of Ti/Gr interface on temperature.

  1. So, I do expect that authors before publications should extensively.

Response: Thanks for your suggestions. We have polished the manuscript and expanded the discussion sections.

In the revised manuscript:

The thermal conductivity of Ti/Gr MMCs is taking into account for TBC effects at Ti/Gr interfaces, and its thermal conductivity increases with graphene volume content. Our findings also provide insights into ways to optimize future thermal management based on MMCs materials.

Reviewer 2 Report

This manuscript studied the thermal conductance at Titanium-Graphene (Ti/Gr) interfacial by using classical molecular dynamics simulation methods. The thermal boundary conductance played an important role in thermal properties of Ti-based metal-matrix composites. They discussed the effects of the size, layer number, temperature, and strain. The change in thermal boundary conductance is also studied and explained by phonon coupling factor and surface potential energy barrier.

The results presented in this work are interesting. Therefore, I would like to recommend this manuscript for publication in Molecules.

There are a few comments on the manuscript:

1. The author would better demonstrate the accuracy of the computed results by systematically comparing their results with the available experimental data.
2. The author studied the influence of layers on TBC. Can we control it accurately in experiment?

3. The author studied the effect of strain on TBC, whether -6% and 6% strain will damage the original structure of the material, resulting in the degradation of other aspects of its properties.
4. It is best to improve English writing.

Author Response

molecules-1554613

Response to editor’s comments

We greatly appreciate all the reviewers for their professional and constructive comments. We have revised our manuscript carefully based on their comments. In the following, we have addressed the reviewer’s comments point by point. It should be noted that the reviewer’s comments are marked in black italic and the corresponding responses are in blue. All the revisions in the revised manuscript are highlighted in red.

Reviewer #2:

This manuscript studied the thermal conductance at Titanium-Graphene (Ti/Gr) interfacial by using classical molecular dynamics simulation methods. The thermal boundary conductance played an important role in thermal properties of Ti-based metal-matrix composites. They discussed the effects of the size, layer number, temperature, and strain. The change in thermal boundary conductance is also studied and explained by phonon coupling factor and surface potential energy barrier.

The results presented in this work are interesting. Therefore, I would like to recommend this manuscript for publication in Molecules.

Response: Thanks for your positive comments.

  1. The author would better demonstrate the accuracy of the computed results by systematically comparing their results with the available experimental data.

Response: Thanks for your suggestion. However, there is no experimental reports about Ti/graphene interfacial thermal conductance studies. We compared our simulation methods with reported Graphene/Cu interfacial thermal conductance results [J. Phys. Condens. Matter 24(24) (2012) 245301], which is about 500 MW/m2K at room temperature. The results using NEMD method which was employed in present work are well consistent with the reported results.

  1. The author studied the influence of layers on TBC. Can we control it accurately in experiment?

Response: Thanks for your comment.

Graphene can be growth on Ti or Ti alloy surface [Surface and Coatings Technology, 401, 126250 (2020); Scientific Reports, 5, 14242 (2015)] or translated on Ti surface physically. The layer number of graphene sheet can be controlled by CVD growth or physical translation. The corresponding statement was supplemented in the manuscript.

In the revised manuscript:

Page 7 line 204: The layer number of graphene sheet can be controlled by CVD growth or physical translation.

  1. It is best to improve English writing.

Response: Thanks for your suggestions. We have polished the manuscript and checked the grammar.
